# Combining Clinical and Genetic Data to Predict Response to Fingolimod Treatment in Relapsing Remitting Multiple Sclerosis Patients: A Precision Medicine Approach

**DOI:** 10.3390/jpm13010122

**Published:** 2023-01-06

**Authors:** Laura Ferrè, Ferdinando Clarelli, Beatrice Pignolet, Elisabetta Mascia, Marco Frasca, Silvia Santoro, Melissa Sorosina, Florence Bucciarelli, Lucia Moiola, Vittorio Martinelli, Giancarlo Comi, Roland Liblau, Massimo Filippi, Giorgio Valentini, Federica Esposito

**Affiliations:** 1Neurology and Neurorehabilitation Unit, IRCCS San Raffaele Hospital, 20132 Milan, Italy; 2Laboratory of Human Genetics of Neurological Disorders, IRCCS San Raffaele Hospital, 20132 Milan, Italy; 3Vita-Salute San Raffaele University, 20132 Milan, Italy; 4Centre Hospitalier Universitaire de Toulouse, CEDEX 9, 31059 Toulouse, France; 5Institut Toulousain des Maladies Infectieuses et Inflammatoires (Infinity), INSERM UMR1291–CNRS UMR5051—Université Toulouse III, CEDEX 3, 31024 Toulouse, France; 6AnacletoLab, Dipartimento di Informatica, Università degli Studi di Milano, 20133 Milan, Italy; 7Data Science Research Center, Università degli Studi di Milano, 20133 Milan, Italy; 8Infolife National Lab, CINI, 00185 Rome, Italy; 9Department of Immunology, Toulouse University Hospitals, CEDEX 3, 31024 Toulouse, France; 10Neuroimaging Research Unit, IRCCS San Raffaele Hospital, 20132 Milan, Italy; 11Neurophisiology Unit, IRCCS San Raffaele Hospital, 20132 Milan, Italy

**Keywords:** multiple sclerosis, fingolimod, machine learning, predictive model, genetic markers, precision medicine

## Abstract

A personalized approach is strongly advocated for treatment selection in Multiple Sclerosis patients due to the high number of available drugs. Machine learning methods proved to be valuable tools in the context of precision medicine. In the present work, we applied machine learning methods to identify a combined clinical and genetic signature of response to fingolimod that could support the prediction of drug response. Two cohorts of fingolimod-treated patients from Italy and France were enrolled and divided into training, validation, and test set. Random forest training and robust feature selection were performed in the first two sets respectively, and the independent test set was used to evaluate model performance. A genetic-only model and a combined clinical–genetic model were obtained. Overall, 381 patients were classified according to the NEDA-3 criterion at 2 years; we identified a genetic model, including 123 SNPs, that was able to predict fingolimod response with an AUROC= 0.65 in the independent test set. When combining clinical data, the model accuracy increased to an AUROC= 0.71. Integrating clinical and genetic data by means of machine learning methods can help in the prediction of response to fingolimod, even though further studies are required to definitely extend this approach to clinical applications

## 1. Introduction

### 1.1. Background

Multiple Sclerosis (MS) is a chronic inflammatory disease of the central nervous system with a complex etiology and a high heterogeneity in terms of clinical presentation and treatment response [1]. In the last decade, the therapeutic opportunities for Relapsing Remitting MS (RRMS) have dramatically expanded, increasing the complexity in disease management and prompting the need for a more individualized approach that takes into account patients’ specific features [2].

### 1.2. Rationale

The existence of genetic factors predisposing to MS has been demonstrated by large genome-wide association studies [3], whereas fewer, smaller studies suggested a possible influence of genetic factors in determining disease severity [4,5,6] and modulating response to first-line treatments like interferon and glatiramer acetate [7,8,9,10,11,12,13]. Fewer data are available on genetic markers associated with response to second-line drugs [14,15] and, to the best of our knowledge, no studies have investigated genetic factors associated with response to fingolimod. 

Fingolimod (FTY) is a highly effective second-line drug approved for RRMS; nonetheless, some subjects show persistent disease activity during FTY treatment and the early detection of these non-responder individuals is essential to promptly address them to more effective therapies [16], given the increasing number of drugs currently available.

Machine learning (ML) algorithms, which can better accommodate the complexities of the relations among variables than the traditional regression methods, are becoming increasingly valuable tools for precision medicine [17,18].

### 1.3. Objective

In the present study, we applied ML methods to investigate the presence of a genetic signature of response to FTY that, in combination with clinical and demographic characteristics, could support the prediction of drug response.

## 2. Materials and Methods

We collected two cohorts of RRMS patients treated with FTY for whom genetic and clinical data during therapy were available. The first cohort (OSR) included 364 patients who started FTY at San Raffaele Hospital MS center in Milan, Italy; the second cohort (CHUT) included 108 FTY-treated RRMS patients from the Centre Hospitalier Universitaire de Toulouse, France.

Baseline and follow-up clinical data during the first 2 years of FTY therapy were obtained; specifically, the following variables were collected: gender, age at onset (AAO), disease duration at treatment start (DD), annualized relapse rate (ARR) in the 2 years prior to FTY start, previous disease modifying treatment, EDSS score at treatment start and at 2 years, number of relapses, and presence of new/enlarging T2-lesions and Gd-enhancing lesions at brain MRI during therapy. Patients treated with natalizumab (NTZ) in the 9 months prior to FTY were excluded due to the known possible “rebound” effect occurring after NTZ discontinuation [19,20] in order to avoid potential misclassification.

Treatment response was assessed at 2 years using the NEDA-3 criterion (No evidence of disease activity [21]), defined as absence of relapses, active MRI lesions, and disease progression. The Ethical Committee at San Raffaele Hospital, Milan, Italy approved the study and all patients signed the informed consent.

### 2.1. Genotyping and Quality Checks

Genetic data for OSR and CHUT cohorts were obtained with the Illumina® HumanOmniExpress Kit. Standard quality control (QC) steps were applied using Plink v1.9beta [22]: per-SNPs QC included removing variants with minor allele frequency (MAF) < 0.01, genotyping rate < 0.97, and deviation from Hardy-Weinberg equilibrium (*p* < 10^−4^). Per-sample QC was performed removing subjects with call rate < 0.95, excess relatedness, sex mismatch, or that were ancestry outliers according to a multidimensional scaling analysis.

The two datasets were merged and SNPs were pruned, discarding those in linkage disequilibrium (LD) (r^2^ > 0.2), in order to remove redundancies in the set of markers, thus reducing the dimensionality of the analyses. Genotypes were coded in discrete additive dosages of minor alleles for ML modeling.

### 2.2. Predictive Models

#### 2.2.1. Genetic Model

In order to avoid selection bias [23] and model overfitting, after combining the two cohorts, we created three subsets called training set (TRset), validation set (Vset), and test set (TEset), including 40%, 40%, and 20% of studied subjects, respectively. The three groups were generated maintaining the proportion between OSR and CHUT cohorts, as well as the EDA/NEDA ratio within each group.

Our preliminary experiments showed that Random Forests (RFs) achieved more stable and accurate results compared to other models (Support Vector Machines, K-nearest-neighbours, decision trees, bagging and boosting) confirming previous ML results on genetic data [24,25]. For this reason, we used RFs to predict response to FTY based on genetic and clinical features.

The TRset was used to train the RFs, whereas the Vset was used to select the group of SNPs (signature) to be included in the predictive model, starting from a dataset of about 113,000 LD-pruned SNPs. Feature selection was performed using a robust cross-validated selection method, based on the stability of the SNPs to be selected. We designed a feature selection algorithm to detect the SNPs that most steadily correlate, according to Pearson coefficient, with the response status across multiple samples of the data. Specifically, we repeated a cross-validation procedure 100 times, storing at each repetition the k top-ranked SNPs in each fold of the cross-validation. Then, the algorithm chose only the SNPs selected with at least a relative frequency f across the repeated cross-validations.

Of note, we used the Vset to select the SNP signature, distinct from both the TRset and TEset, thus avoiding the selection bias and reducing the overfitting of the trained model [23]. Finally, the TEset was used to test the models trained using the selected SNP signature.

We performed an extensive model selection by comparing the results of about 2000 models resulting from different combinations of the learning parameters of the RFs by varying the number of decision trees of the ensemble between 10 and 100, the maximum number of nodes for each decision tree from 1 to 100, the minimum number of examples stored in each leaf node from 1 to 10, and of the robust cross-validated feature selection method by varying the minimum relative frequency f of each feature between 0.05 and 1, and the number k of top features to be considered at each iteration of the cross-validation between 50 and 1000. It is easy to see that for a fixed k, large values of f (close to 1) lead to small sets f of selected SNPs, whereas small values lead to larger SNP signatures.

The best models were selected and evaluated on the TEset as measured by the area under the receiver operating characteristic curve (AUROC); the Area Under the Precision-Recall Curve (AUPRC), F-score, and accuracy were also calculated to evaluate models performance.

The analyses were performed using R statistical environment, version 3.6.3; specifically, the “randomForest”, “caret” and “precrec” R packages were used, as well as a set of functions designed specifically for the present analyses that are available upon request.

#### 2.2.2. Combined Clinical and Genetic Model

As for the genetic model, we trained the RFs separately on the clinical data and selected four signatures through which we achieved the best results on the Vset, starting from 17 clinical parameters (gender, AAO, age at treatment start, DD, ARR in the 2 years before FTY, baseline EDSS, presence of new T2-lesions and Gd+-lesions at baseline MRI plus other nine binary features derived from the categorization of previous treatments). We ran about 2000 different models and selected the top four models according to their performance.

Finally, we combined the genetic and clinical models by means of multi-view random forests, where each decision tree was separately trained on a different bootstrap sample of either genetic or clinical data. The predictions of the resulting decision trees were finally combined to obtain the consensus multi-view prediction of the RF ensemble [26].

#### 2.2.3. Predictive Performance of the Genetic Model in Patients Treated with Other Immunomodulatory Drugs

In order to assess whether the set of SNPs predicted as associated to a better response were rather markers of a mild disease activity, we considered two independent cohorts of patients treated with glatiramer acetate (GA, n = 273) and beta-interferons (IFN, n = 304), enrolled at San Raffaele Hospital MS center, with available genetic data. For these subjects, the same clinical information previously mentioned was collected during treatment, and response was assessed at 2 years using the NEDA-3 criterion. The genetic model selected in FTY-treated patients was applied to GA and IFN-treated subjects, and the AUROC was calculated to assess whether it was specific for FTY response or if it was more generally associated to disease activity.

#### 2.2.4. Classification Performance of the Combined Model

In order to test the ability of the selected model to identify patients that respond to FTY, we considered the predicted probability of non-response to FTY as calculated on the independent TEset by the selected clinical and genetic model, and we divided these values into tertiles; the highest tertile included the patients that the model predicted with higher likelihood as being non-responders to FTY (PrNR), whereas the lowest tertile grouped patients more likely to respond to treatment (PrR). We then compared the lowest and highest tertiles in terms of disease activity during FTY treatment; specifically, we compared the number of new and/or enlarging T2 lesions and the number of relapses between the two groups by means of the non parametric Mann–Whitney test; we then compared the proportion of patients free from MRI activity and free from clinical activity and the proportion of patients achieving the NEDA outcome by means of a chi-square test.

## 3. Results

### 3.1. Summary of Results

In the present study we identified a genetic signature of 123 SNPs that was able to predict FTY response with an area under the receiver operating characteristic (AUROC) = 0.65 and the model accuracy further increased when considering also clinical data (AUROC = 0.71). Even though the model predictivity is not enough for implementation in clinical practice, our findings suggest that the combination of clinical and genetic data by means of machine learning methods can support in the prediction of response to FTY.

### 3.2. Detailed Results

We applied the genetic and the combined clinical–genetic model to the prediction of the EDA/NEDA status using the OSR/CHUT combined cohort according to the experimental set-up described in Section 2.

More precisiely, 342 FTY-treated RRMS patients from the OSR cohort and 78 from the CHUT dataset were considered for the following steps, after removal of patients treated with NTZ in the previous nine months and genetic QCs.

Among them, 17 and 22 patients in the CHUT and OSR cohort, respectively, were excluded because we were not able to correctly classify them according to the NEDA-3 criterion due to missing clinical data.

The final analysis was performed on 381 patients, of whom 197 showed evidence of disease activity (EDA) and 184 were NEDA during the two-year follow-up. Among them, 152 entered the TRset, 152 the Vset, and 77 the independent TEset. Clinical and demographic characteristics of the included patients, stratified according to the three sets, are described in Table 1.

### 3.3. Genetic Model

We applied the robust cross-validation procedure introduced in Section 2.2.1 to select the genotypic signatures predictive of the EDA/NEDA status. Using the training dataset only to avoid the selection bias, we selected the *top-f* SNPs most steadily correlated with the response status across ten-fold cross-validation repeated 100 times (Table 2). For instance, in Table 2, the g2 model includes a signature with 123 SNPs (*sign*) selected using the first 500 top-ranked features (*top-f*) with a frequency (*min-fr)* equal to 0.1 (i.e., SNPs selected at least 10% of times across the repeated cross-validation procedure). Random forests were trained on the training set using the SNP signatures, and their parameters were selected by evaluating their AUROC, AUPRC, and F-score performance on the independent validation set to limit overfitting (see Table 2 for details). Finally, the generalization performance of the best models was evaluated on the independent test set (Table 2).

The AUROC calculated on the independent TEset was close to 0.65, and F-scores and AUPRC were even higher, suggesting that the NEDA classification can be learnt from genetic data, although the accuracy of the prediction was not enough for application in everyday clinical practice.

Due to the smaller number of variants included in the second model (g2)—that makes it more appealing for a potential clinical application—and the very similar predictive performance, we selected it for further analyses; the list of the 123 SNPs prioritized by the analysis is reported in Appendix A.

In order to gain some biological insight from the detected signature associated with the FTY response, we performed an enrichment analysis to identify the pathways in which the selected variants were involved. First, we annotated the variants using the Ensembl Variant Effect Predictor tool (VeP), then, we selected genes for which at least a moderate impact was predicted and belonging to the biotypes “protein coding”, “processed transcript”, and “retained intron”. We then performed an over representation analysis with the tool Webgestalt [27] using the KEGG pathway database as reference. Of the 73 annotated genes, 30 were annotated to functional categories present in the database and, though no pathway survived multiple testing correction, among the top enriched pathways (Table 3) we found “Sphingolipid signaling pathway” (*p*-value: 0.008), “Sphingolipid metabolism” (*p*-value: 0.011), “Cell adhesion molecules (CAMs)” (*p*-value: 0.013), and “Inflammatory bowel disease (IBD)” (*p*-value: 0.021). These results suggest that our algorithm is indeed able to identify genes that play a role in modulating FTY mechanism of action and therapeutic response.

### 3.4. Combined Clinical and Genetic Model

We then applied the same algorithm to the clinical data: interestingly, among the clinical features steadily selected for inclusion in the model, there were age at treatment start, ARR in the 2 years before FTY start, and presence of new T2 and Gd+ lesions at baseline, which are already known to be associated with disease activity upon FTY treatment [19].

Table 4 reports the four best models trained on the clinical data that performed slightly better in terms of AUROC (0.69) and AUPRC (0.67), compared to the genetic model.

We then combined the selected genotypic and clinical signatures using multi-view random forests and obtained an increase in the predictive accuracy of the model, with an AUROC of 0.71 and an AUPRC of 0.73 on the TEset (Table 5).

These results suggest that combining clinical and genetic data can help in predicting response to FTY.

To further confirm this evidence, we considered the predicted probability of non-response to FTY as calculated on the independent TEset by the combined model g2-c1, and we considered the patients that the model predicted with higher likelihood as being non-responders to FTY (PrNR) and those deemed more likely to respond to treatment (PrR).

As expected, a significantly greater proportion of patients predicted with higher likelihood as being non-responders had evidence of disease activity during the 2-year follow-up, compared to the patients predicted with higher likelihood as being responders (75% vs. 27%, *p*: 0.0019, Figure 1A). Moreover, PrNR patients also showed a higher neuroradiological activity during FTY treatment, both in terms of number of active lesions (2.1 vs. 0.61 on average in the PrNR and PrR group, respectively, *p*: 0.034 Figure 1B) and of proportion of individuals with an active MRI scan (50% vs. 11.5% *p*: 0.005, Figure 1C). The same direction of effect was also found when considering the level of clinical activity, PrNR patients showing a higher number of relapses (0.25 vs. 0.15 *p*: 0.40, Figure 1D) and a larger proportion of patients with a clinical reactivation (20.8% vs. 7.7% *p*: 0.24, Figure 1E) compared to PrR, although not statistically significant.

### 3.5. Evaluation of the Model in Independent Cohorts of Patients Treated with First-Line Drugs

Finally, to test whether our genetic model was specific for predicting FTY treatment response or was a prognostic algorithm that more broadly predicts MS disease activity, we tested it in patients treated with IFN and GA. In the IFN cohort, 110 patients had NEDA and 189 EDA, whereas in the GA cohort, 93 had NEDA and 170 EDA. We then tested the model trained on the FTY cohort on these 2 datasets and obtained an AUROC of 0.55 and 0.51, respectively, suggesting that the model holds specificity for FTY therapy and is not able to predict response to IFN and GA. We did not test the clinical model on these cohorts, given that the identified clinical predictors are mainly prognostic factors already known to be associated to response to first-line treatment, so we do not expect the clinical model to be specific to FTY-treated patients.

## 4. Discussion

MS is a complex disorder, with substantial heterogeneity in terms of response to treatments, that would greatly benefit from a more individualized management; indeed, predicting the response of MS patients to therapies has been an open problem for many years [28]. Recently, due to the advancements and the spreading of artificial intelligence (AI) methodologies, several ML algorithms have been applied in the MS field; in particular, AI has been applied to MRI data to perform lesion and tissue segmentation [29,30] and to aid in the differential diagnosis with MS mimics [31,32,33]. A few studies also applied AI to clinical data in order to predict disease course and progression [34,35] but, to our knowledge, a single study applied ML methods to genetic data in order to predict treatment response in patients treated with a first-line drug [36].

In the present study, we applied a supervised ML algorithm to clinical and genetic data in a cohort of MS patients treated with FTY in order to develop a predictive model of response to the drug, and we identified a genetic signature of 123 SNPs that was able to predict treatment response in an independent cohort, even though its predictive accuracy is not enough for its use in clinical practice (AUROC = 0.65, AUPRC = 0.66). When considering clinical data only, a similar performance was obtained (AUROC = 0.69, AUPRC = 0.67), suggesting that clinical information can possibly add to the predictive value of the model. Interestingly, among the clinical parameters that were retained by the robust feature selection algorithm were the age at treatment start, the ARR in the 2 years before FTY, and the presence of new T2 and Gd+ lesions at baseline MRI, variables that have already been associated with disease activity and response to FTY treatment [19]. These results indicate that, even if the predictive power of the model is limited, our ML approach is able to identify clinically significant predictors.

Similarly, when applying an over-representation analysis to the genetic hits selected by the algorithm, we found that there was an enrichment of genes implicated in sphingolipid metabolism and in cell-adhesion pathways, further supporting the ability of the model to identify biologically meaningful signals implicated in response to FTY.

As expected, the highest accuracies were obtained when considering both clinical and genetic data: in fact, the combined model yielded an AUC of 0.71 and an AUPRC of 0.73. Such predictive performance, although insufficient to guide decision making in clinical practice, suggests that ML methods have the potential to stratify patients for whom FTY is a treatment option. Moreover, our results are in line, if not better, than those obtained by a previous study on glatiramer acetate [36].

When analyzing only patients whose response state is predicted with higher or lower likelihood as being non-responder to FTY, the algorithm was able to identify two categories of patients that significantly differ in terms of disease activity during FTY treatment: only 25% of PrR patients showed disease reactivation in the 2 years after treatment start compared to 75% of PrNR. Similarly, PrNR patients have a significantly greater rate of MRI activity and showed a modest trend towards more relapses compared to PrR.

Among the strengths of our work is the detailed clinical characterization that allowed the improvement of the genetic-only model. Most importantly, our study design included a validation step in an independent cohort that was completely blind to the model training and to the features selection process; compared to internal cross-validation procedures, the availability of an external independent population allows for an unbiased and not overly-optimistic estimation of the generalizability of the model and reduces the risk of model overfitting, thus increasing the reliability of our findings [23,37,38].

Moreover, our robust procedure to select informative SNP signatures, combined with an unbiased evaluation of the experimental results using an integrated ML approach, offers a methodological framework that can be applied to MS or also other complex human traits.

On the other hand, we are aware of some limitations of the present study, mainly due to the modest sample size, especially in relation to the high dimensionality of genetic data [39], with an independent test set of only 77 patients, and the relatively short follow-up.

## 5. Conclusions

Our study shows that ML methodologies hold potential to be applied to clinical and genetic data towards a more personalized approach in MS and advocates for further studies addressing the issue of predicting treatment response. In perspective, the availability of larger cohorts and an effective combination of clinical, omics, and other types of data (e.g., imaging data) through an integrated ML approach [40] (such as our proposed multi-view RF ensembles) could lead to improved response predictions potentially applicable in clinical practice.

## Figures and Tables

**Figure 1 jpm-13-00122-f001:**
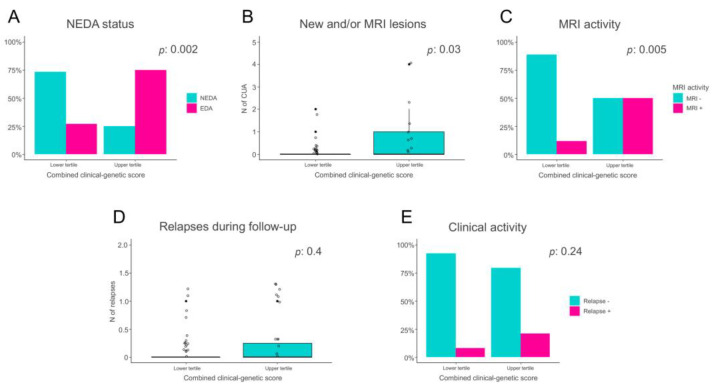
Comparison of disease activity levels between patients predicted to be non-responders to FTY (PrNR) and patients likely to respond to treatment (PrR). (**A**) Proportion of patients with No Evidence of Disease Activity (NEDA) and Evidence of Disease Activity (EDA) in the PrR and PrNR groups. (**B**) Number of new and/or active lesions in the PrR and PrNR groups. (**C**) Proportion of patients with or without MRI activity in the PrR and PrNR groups. (**D**) Number of clinical relapses in the PrR and PrNR groups. (**E**) Proportion of patients with or without clinical activity in the PrR and PrNR groups.

**Table 1 jpm-13-00122-t001:** Clinical and demographic characteristics of included patients, stratified according to TRset, Vset, and TEset.

	Whole Cohort (n: 381)	TRset (n: 152)	Vset (n: 152)	TEset (n: 77)	*p*-Value
F:M ratio	270:111	101:51	119:33	50:27	0.03
Age at disease onset, mean ± SD	29 ± 9.5	28.3 ± 9.2	29.1 ± 9.7	30.2 ± 9.6	n.s.
Age at FTY start, mean ± SD	39.5 ± 9.5	38.8 ± 9.1	39.6 ± 9.7	40.4 ± 9.8	n.s.
Disease duration (yrs), mean ± SD	10.5 ± 7.6	10.5 ± 7.1	10.5 ± 8.2	10.2 ± 7.1	n.s.
ARR in the 2 years prior FTY, mean ±SD	0.82 ± 0.84	0.74 ± 0.83	0.93 ± 0.90	0.78 ± 0.73	n.s.
Previous DMT					n.s.
Naïve	30 (7.9%)	12 (7.9%)	13 (8.6%)	5 (6.5%)	
No therapy	26 (6.8%)	12 (7.9%)	10 (6.6%)	4 (5.2%)	
IFN	149 (39.1%)	59 (38.8%)	58 (38.2%)	32 (41.5%)	
GA	104 (27.3%)	40 (26.3%)	40 (26.3%)	24 (31.2%)	
DMF	13 (3.4%)	4 (2.6%)	6 (3.9%)	3 (3.9%)	
Teriflunomide	11 (2.9%)	4 (2.6%)	3 (2%)	4 (5.2%)	
Immunosuppressants	17 (4.5%)	6 (4.0%)	9 (5.8%)	2 (2.6%)	
Natalizumab	29 (7.6%)	13 (8.6%)	13 (8.6%)	3 (3.9%)	
Other	2 (0.5%)	2 (1.3%)	0 (0%)	0 (0%)	
EDSS at FTY start, median (range)	2.0 (0–7.0)	2 (0–6.0)	2 (0–7.0)	2 (0–6.0)	n.s.
<Patients with Gd+ lesions at baseline brain MRI scan	33.5%	35.1%	31.7%	33.8%	n.s.
Patients with new/enlarged T2 lesions at baseline brain MRI scan	49.1%	45.7%	54.5%	44.9%	n.s.

FTY: fingolimod; ARR: annualized relapse rate; DMT: disease modifying treatment; IFN: interferon; GA: glatiramer acetate; DMF: dimethyl fumarate; Gd+ lesions: gadolinium enhancing lesions. P-value refers to the comparison between the 3 sets.

**Table 2 jpm-13-00122-t002:** Best random forest models trained on the genotypic data.

Model	Top-f	Min-fr	Sign	Ntree	Nodesize	Maxn	TR AUROC	TR AUPRC	TR F	TR Acc	TE AUROC	TE AUPRC	TE F	TE Acc
g1	500	0.05	1022	20	10	10	0.8493	0.8476	0.7973	0.796	0.65	0.6483	0.6837	0.5194
g2	500	0.1	123	10	10	30	0.8438	0.8496	0.7861	0.7565	0.6446	0.663	0.7142	0.5844
g3	500	0.05	1022	10	1	15	0.838	0.8671	0.7567	0.7631	0.5801	0.5907	0.745	0.6623
g4	500	0.15	8	10	2	60	0.858	0.8712	0.8	0.7828	0.6135	0.6176	0.7102	0.5974

The table shows the four best genetic models, named g1, g2, g3, and g4. Parameters of the feature selection method: top-f stands for the number of the top-ranked selected features; min-fr is the minimum frequency of the feature to be selected, sign is the number of selected features. Parameters of the random forests: ntree is the number of trees; nodesize is the minimum size of each leaf of the tree; maxn is the maximum number of nodes for each tree. TR_AUROC, TR_AUPRC, TR_F, TR_acc, TE_AUROC, TE_AUPRC, TE_F, TE_acc stand for AUROC, AUPRC, F-score, and Accuracy, respectively, on the training and test set.

**Table 3 jpm-13-00122-t003:** Over-representation analysis of the genes selected by the ML model.

Description	Size	Expected	Enrichment	*p* Value	FDR
Renin secretion	65	0.23	13.26	0.001	0.42
Calcium signaling pathway	183	0.64	6.28	0.003	0.51
Sphingolipid signaling pathway	118	0.41	7.30	0.008	0.65
Sphingolipid metabolism	47	0.16	12.22	0.011	0.65
Cholesterol metabolism	50	0.17	11.49	0.013	0.65
Cell adhesion molecules (CAMs)	144	0.50	5.98	0.013	0.65
cGMP-PKG signaling pathway	163	0.57	5.29	0.018	0.67
Cortisol synthesis and secretion	64	0.22	8.98	0.021	0.67
Inflammatory bowel disease (IBD)	65	0.23	8.84	0.021	0.67
Long-term potentiation	67	0.23	8.58	0.022	0.67

Size: gene set size; Expected: n° of predicted overlap; Enrichment ratio: ratio between the observed and expected overlap; FDR: false discovery rate.

**Table 4 jpm-13-00122-t004:** Best Random forest models trained on the clinical data.

Model	Top-f	Min-fr	Sign	Mtry	N Tree	Node Size	Maxn	TR AUROC	TR AUPRC	TR F	TR acc	TE AUROC	TE AUPRC	TE F	TE acc
c1	8	0.5	9	10	100	2	100	1	1	1	1	0.6895	0.6709	0.7339	0.6494
c2	8	0.5	9	4	20	2	100	0.9785	0.9803	0.9255	0.9211	0.6405	0.7320	0.7091	0.6364
c3	8	0.5	9	10	20	2	30	0.9152	0.9281	0.8434	0.8289	0.623	0.6422	0.7379	0.6494
c4	2	0.05	14	3	10	1	100	0.9971	0.9974	0.9684	1	0.6895	0.6709	0.7339	0.6494

The table shows the four best clinical models, named c1, c2, c3 and c4. Parameters of the feature selection method: top-f stands for the number of the k top-ranked selected features; min-fr is the minimum frequency f of the features to be selected. Sign is the cardinality of the clinical signature. Parameters of the random forests: mtry is the number of the randomly selected features at each node of the decision trees, ntree is the number of trees; nodesize is the minimum size of each leaf of the tree; maxn is the maximum number of nodes for each tree. TR_AUROC, TR_AUPRC, TR_F, TR_acc, TE_AUROC, TE_AUPRC, TE_F, TE_acc stand for AUROC, AUPRC, F-score, and Accuracy, respectively, on the training and test set.

**Table 5 jpm-13-00122-t005:** Best multi-view random forest model trained on combined clinical and genetic data.

Model	TR_AUROC	TR_AUPRC	TR_F	TR_Acc	TE_AUROC	TE_AUPRC	TE_F	TE_acc
g2-c1	0.9997	0.9997	0.9933	0.9934	0.7095	0.7328	0.7328	0.6623

The model name g2-c1 refers to the single-layer models it derives from. TR_AUROC, TR_AUPRC, TR_F, TR_acc, TE_AUROC, TE_AUPRC, TE_F, TE_Acc stand for AUROC, AUPRC, F-score, and Accuracy, respectively, on the training and test set.

## Data Availability

Patient-level data presented in this study are available on request to the corresponding author. Data regarding genetic variants selected by the model are included in Appendix A.

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
