# Peer review of "Combining Clinical and Genetic Data to Predict Response to Fingolimod Treatment in Relapsing Remitting Multiple Sclerosis Patients: A Precision Medicine Approach"

_jpm, 2023, doi:10.3390/jpm13010122_

Round 1

Reviewer 1 Report

The case study submitted by Ferra et al. is well written article and scientifically valid. I have no problem with the content and rigor of the experiments performed. But the statistical reporting can be improved. It would be nice if the authors can explicitly mention about the statistical tests that have been performed for each analysis. A separate section/table will be beneficial and highly recommended. 

Reviewer 2 Report

Introduction:

Line 56: spacing before (ML)

Line 62: mention the full form before mentioning abbreviation for AUROC

Important comment for lines 58-66: I do not see any rationale of the study; separate the aims/objectives and rationale into a separate paragraph and please explain to the reader what this study seeks to do. It is unclear at this moment.

Methods:

Are overall good; even if an ethical approval is not required, state so explicitly.

Results:

Lines 168-172: these need to reordered. The results need to be made clearer. A summary paragraph of 5-8 lines is required at the start of the results where I can understand what you found on the get go.

Lines 186-188: add more details. Simply writing that table 2 shows parameters is insufficient.

Line 214 and everywhere else: mention p-value as “p-value=”

Line 260: is clumsy; please reword.

Figure 1: is blurry; please upload a better quality image.

Discussion:

Line 295-296: Cite the following study: Taj HM, Talib M, Siddiqa S, et al. What Do We Know So Far about Ofatumumab for Relapsing Multiple Sclerosis? A Meta-Analytical Study. Healthcare (Basel). 2022;10(11):2199. Published 2022 Nov 2. doi:10.3390/healthcare10112199

The discussion is not at par with MDPI standards. After line 337, you come to strengths of your work. What you are suggested to do is add a couple of more paragraphs, review the literature, add more current citations and revamp it.

Conclusion:

This paper needs a conclusion paragraph. It is seeming very confusing without one.

Round 2

Reviewer 2 Report

Good to go now.